# Syntactic and Semantic Influences on the Time Course of Relative Clause Processing: The Role of Language Dominance

**DOI:** 10.3390/brainsci11080989

**Published:** 2021-07-27

**Authors:** Michael C. Stern, LeeAnn Stover, Ernesto Guerra, Gita Martohardjono

**Affiliations:** 1Linguistics Department, Yale University, 370 Temple St, New Haven, CT 06511, USA; 2Linguistics Program, The Graduate Center, City University of New York, 365 Fifth Ave, New York, NY 10016, USA; lstover@gradcenter.cuny.edu (L.S.); gmartohardjono@gc.cuny.edu (G.M.); 3Center for Advanced Research in Education, Institute of Education, Universidad de Chile, Periodista José Carrasco Tapia 75, Santiago de Chile 7550000, Chile; ernesto.guerra@ciae.uchile.cl

**Keywords:** language dominance, relative clause processing, bilingualism, individual differences, visual world paradigm, prediction, semantic processing, syntactic processing

## Abstract

We conducted a visual world eye-tracking experiment with highly proficient Spanish-English bilingual adults to investigate the effects of relative language dominance, operationalized as a continuous, multidimensional variable, on the time course of relative clause processing in the first-learned language, Spanish. We found that participants exhibited two distinct processing preferences: a semantically driven preference to assign agency to referents of lexically animate noun phrases and a syntactically driven preference to interpret relative clauses as subject-extracted. Spanish dominance was found to exert a distinct influence on each of these preferences, gradiently attenuating the semantic preference while gradiently exaggerating the syntactic preference. While these results might be attributable to particular properties of Spanish and English, they also suggest a possible generalization that greater dominance in a language increases reliance on language-specific syntactic processing strategies while correspondingly decreasing reliance on more domain-general semantic processing strategies.

## 1. Introduction

Relative clauses have long offered a window through which to understand language processing, as they entail a long-distance dependency between a noun phrase (NP) “filler” and an unpronounced “gap”. This filler-gap dependency induces a well-attested processing cost [1], which has been found to be greater for object-extracted relative clauses (ORCs), such as (1), than subject-extracted relative clauses (SRCs), such as (2), across a range of languages (including Spanish [2] and English [3]) and experimental methodologies such as pupillometry [4,5], eye-tracking while reading [2,3], visual world eye-tracking [6,7,8], event-related potentials (ERP: [9]), positron emission tomography (PET: [10,11]), and functional magnetic resonance imaging (fMRI: [12,13,14]).

(1)The monkey_i_ [that the rabbit bites ____i_] grabs the cat.(2)The monkey_j_ [that ____j_ bites the rabbit] grabs the cat.

A number of hypotheses have been proposed to explain this processing asymmetry, or SRC preference (see [15] for a review). One group of hypotheses attributes the asymmetry to the difference in syntactic structure between SRCs and ORCs. For example, ORCs have been argued to be more structurally complex than SRCs, increasing processing demands [16,17,18,19]. In addition, constraints on working memory have been proposed to drive the parser to resolve dependencies as soon as possible, favoring SRC interpretations (with an earlier gap) over ORC interpretations (with a later gap) [20,21,22,23]. Note, however, that this account makes different predictions for languages in which the distance between the filler and the gap is shorter in ORCs than SRCs, e.g., Mandarin [24]. On the other hand, semantic influences on the SRC preference have also been discovered, casting doubt on the exclusively syntactic nature of the pattern. For example, the SRC preference has been shown to be modulated, even to the point of disappearance, by varying the lexical animacy of the NPs in the sentence [25,26,27]. In general, the difficulty of ORC processing is attenuated when the filler NP is lexically inanimate, compared to when it is lexically animate. In addition, the results of one study [8] suggest a role for the *perceptual* (i.e., context-specific) animacy of lexically inanimate NPs. To the authors’ knowledge, no studies have yet investigated whether a similar context-specific manipulation affects relative clause processing with lexically *animate* NPs. However, research on the cognition of event construction has attested to the ability of perceivers to distinguish animate *agents* from animate *patients* in a very short time and to a preference for assigning agency over patiency to characters in an event [28,29,30,31]. This “agent advantage” has been shown to manifest in a wide variety of human behavior, from online sentence processing in adults to comprehension in child first-language (L1) acquisition to gestural production in emerging sign languages (for a review, see [30]). It might be expected, therefore, that the context-specific agency of animate NPs would affect relative clause processing analogously to the effects of lexical and perceptual animacy.

So far, the majority of research on the SRC processing preference has focused on monolingual native speakers [15]. However, a growing body of work suggests that the preference tends to extend to second-language (L2) processing in learners (e.g., L1 Spanish-L2 English [22]; L1 German-L2 Dutch [32]). Offline comprehension and production studies also converge on an SRC preference in both the L1 and L2 of bilinguals (e.g., Spanish-Basque bilinguals [33]; Russian-English bilinguals [34]). While some studies have demonstrated an effect of L2 exposure on parsing preferences during comprehension of syntactically ambiguous relative clauses in the L1 (e.g., in L1 Spanish-L2 English bilinguals [35]), relatively few studies have investigated the effects of bilingualism on the SRC preference in L1 processing.

In a previous study from our lab, Stern and colleagues [6] used eye-tracking in the visual world paradigm to examine the SRC processing preference in Spanish-English bilinguals’ L1, Spanish. All participants were fluent in both Spanish and English, but participants were divided into two groups based on their history of language exposure. Those who were born in the anglophone U.S. or moved there before age eight were termed “heritage bilinguals”. Heritage bilinguals are usually defined as bilinguals who grew up using a language in the home that differed from the dominant community language (see, for example, [36]). Participants who moved to the anglophone U.S. at age 17 or older were termed “late bilinguals”. This study [6] found that the late bilingual group demonstrated the expected SRC preference, evidenced by increased fixations to the target image during SRCs compared to ORCs, while the heritage bilingual group demonstrated basically equivalent processing speed during SRCs and ORCs, i.e., no SRC preference. The late bilingual group demonstrated faster processing speeds than the heritage bilingual group during SRCs, and interestingly, the late bilingual group also demonstrated *slower* processing of ORCs than the heritage bilingual group. Recall that the SRC processing preference has previously been observed in monolingual speakers of both Spanish [2] and English [3]. Moreover, the lexical items used in this study were highly frequent. Therefore, this group-level processing difference cannot be explained as the outcome of cross-linguistic influence or differences in speed of lexical access. Rather, in order to explain this pattern of results, it was proposed that the heritage bilinguals’ reduced quantity of Spanish input and use caused a general reduction in their predictive processing in Spanish, which slowed down processing during SRCs (the canonical or expected structure) but also reduced the processing cost incurred by encountering ORCs (the less expected structure). Thus, according to this proposal, the heritage bilingual group’s reduced predictive processing compared to the late bilingual group corresponded with both disadvantages and advantages with regard to processing time. However, as a result of the broad criteria used to group participants in that study, the precise causes of this group-level processing difference remained largely unknown.

In a follow-up study using the same experimental paradigm, Stover and colleagues [7] recruited an additional group of Spanish-English bilinguals who had moved to the anglophone U.S. between the ages of 10 and 16. These participants did not fit clearly into either of the traditional categories of “heritage bilingual” or “late bilingual”; rather, they bridged the gap of language history between these two groups. In addition, rather than dividing participants into groups based on the age of arrival, the follow-up study [7] conducted an individual-level analysis using a gradient, multidimensional measure of *language dominance* (see Section 2.1 [37]). The results of that study largely supported the findings of [6] but extended them to the level of the individual: greater Spanish dominance was found to gradiently increase SRC processing speed and gradiently decrease ORC processing speed in a basically linear fashion. Interestingly, the negative effect of dominance on ORC processing speed was more pronounced than the positive effect of dominance on SRC processing speed, suggesting that the processing cost incurred during ORCs actually outweighs the advantage afforded during SRCs. In this way, [7] demonstrated the utility of multidimensional, individual-level measures in shedding light on bilingual language processing.

However, in [6,7], fixation proportions were binned across relatively large time windows, so those studies did not allow fine-grained examinations of the *time course* of processing or the way that the effects of language dominance might have varied over the course of processing. This constitutes a particularly important gap with regard to the role of prediction: in order to isolate the effects of prediction during processing, it is crucial to obtain measures *before* the onset of relevant linguistic information [38]. Since fixation proportions in those two previous studies were binned across the entire duration of the relative clause, the precise temporal onset of the SRC preference remains undetermined, as does, consequently, the role of predictive processing. Additionally, [6,7] only examined fixations to the target image, ignoring fixations to the two competitor images. However, as will be described in Section 2.2, fixations to each competitor image indicate particular processing preferences at particular moments during the linguistic stimulus. Therefore, examining fixations to the competitor images, in addition to the target image, has the potential to shed unique light on participants’ time-dependent processing strategies throughout the linguistic stimulus.

In the present study, we used the same experimental paradigm as the two studies described above but treated both language dominance and time as continuous variables in order to gain a finer-grained understanding of the relationship between individual-level language dominance and the time course of processing. An important benefit of continuous monitoring (vs. comprehension questions, or aggregated time bins) is that the gaze record provides insights into how comprehension processes unfold over time on a scale of milliseconds and how the immediate visual context contributes to spoken sentence comprehension [39]. In particular, we examined fixations *before* the onset of the relative clause in order to examine processing preferences in the absence of disambiguating linguistic information, i.e., predictive processing. In addition to manipulating the syntactic structure of the linguistic stimuli (SRC vs. ORC), we also investigated the effects of differences in the depicted *agency* of the referents of the NPs in the visual stimuli. While all of the NPs in our stimuli were lexically animate, they varied in how many actions they were performing in each image (zero, one, or two). Our primary research questions can be stated as follows:Do bilingual listeners demonstrate a syntactic preference for SRCs over ORCs during L1 processing?Do bilingual listeners demonstrate a semantic preference to interpret animate NPs as agents rather than patients during L1 processing?Does bilingual language dominance, operationalized as a continuous, individual-level variable, affect the time course of either of these processing preferences?

Based on the studies reviewed above, we expect to observe both a syntactic preference for SRCs and a semantic preference for agency. Moreover, we expect that greater L1 dominance will gradiently exaggerate the syntactic preference for SRCs. While at least one study has found that animacy effects are diminished in L2 RC processing [40], to our knowledge, no studies have yet examined the influence of bilingualism on the agency preference in event cognition. Therefore, we have no specific predictions regarding the effects of language dominance on the agency preference, so this aspect of our analysis is largely exploratory at this point.

## 2. Materials and Methods

### 2.1. Participants

We recruited 56 Spanish-English bilingual adults (aged 19–55: *M* = 27.786, *SD* = 9.103; 39 women, 17 men) who lived in New York City at the time of testing. Before completing the main language dominance questionnaire (described in the following subsection), participants completed a short questionnaire to determine if they met the study’s inclusion criteria. We only included participants who self-reported that they had normal or corrected-to-normal vision and hearing, that they were fluent in both Spanish and English as indicated by a score of at least three on a five-point Likert scale, and that Spanish was their primary language of communication with their caregivers until at least age 10. Most participants (*n* = 45) were born outside of the anglophone U.S., and the age of arrival ranged from 2 to 45 years of age (*M* = 16.913, *SD* = 9.335). This wide range in our sample’s language history was expected to lead to a broad distribution of individual language dominance scores, which is beneficial for assessing the effects of language dominance as a continuous variable.

#### Language Dominance Score

In order to operationalize language dominance as a gradient individual measure, we used the composite dominance score generated from responses on the Bilingual Language Profile (BLP: [37]). The BLP is a questionnaire developed to assess a bilingual individual’s language dominance, taking into account dominance’s gradience, relativity, multidimensionality, context-dependence, and time-dependence ([41]; see also [42,43,44]). To this end, questionnaire items were initially drawn from existing bilingual questionnaires such as the Language Experience and Proficiency Questionnaire (LEAP-Q: [45,46]), the Bilingual Dominance Scale (BDS: [47]), and the Self-Report Classification Tool (SRCT: [48]). Questions were refined for clarity and succinctness based on pilot testing and then validated with responses from 68 English-French bilinguals. Based on factor analyses, items were grouped into four components or “modules”: history, use, proficiency, and attitudes. The “history” module comprises six questions probing age of acquisition and length of exposure, the five questions in the “use” module elicit percentages of current use in both languages, and the remaining eight questions for the “proficiency” and “attitudes” modules employ six-point Likert scales. Average scores from the four modules are combined to generate a composite score of relative language dominance for each respondent ranging from −218 to 218, with positive scores indicating relative dominance in Spanish, negative scores indicating dominance in English, and a score of zero indicating balanced bilingualism (see [37] for a description of the procedure for calculating the composite dominance score).

A histogram of participants’ dominance scores is presented in Figure 1, and a summary of participant characteristics is presented in Table 1 and Table 2. For the sake of illustration, the participant data in Table 1 and Table 2 is divided into two groups: “Spanish-dominant” (positive dominance score) and “English-dominant” (negative dominance score). However, it is important to stress the continuous nature of the language dominance score and the corresponding lack of a group variable in any of our inferential analyses. Our sample included a wide range of dominance scores, ranging from −104.980 (very English-dominant) to 148.838 (very Spanish-dominant). The mean is slightly above zero (21.121). A Shapiro–Wilk normality test indicates that the distribution is approximately normal (W = 0.977, *p* = 0.384), and a skewness value of −0.151 indicates that the distribution is approximately symmetric. There were strong correlations between dominance and age of arrival to the anglophone U.S. (Spearman’s *ρ* = 0.777, *p* < 0.001), and between dominance and length of residence in the anglophone U.S. (*ρ* = −0.761, *p* < 0.001). With the knowledge that reliable measures of language dominance have been shown to correlate with age of arrival [49,50] and length of residence [51], combined with separate evidence of the validity of the BLP [41], we assume that these relative dominance scores are a reliable representation of the individuals in our sample. Lastly, there was no correlation between dominance and age at the time of testing (*ρ* = −0.093, *p* = 0.495), suggesting that any effects of dominance on processing observed in the present study cannot be attributed to confounding effects of age [52].

### 2.2. Stimuli and Design: Eye-Tracking Experiment

The linguistic materials consisted of 40 complex Spanish sentences, including 10 experimental sentences with subject relative clauses (SRCs) and 10 experimental sentences with object relative clauses (ORCs), plus 20 filler sentences (see Figure A6, Figure A7, Figure A8, Figure A9, Figure A10, Figure A11, Figure A12, Figure A13, Figure A14, Figure A15, Figure A16, Figure A17, Figure A18, Figure A19, Figure A20, Figure A21, Figure A22, Figure A23, Figure A24, Figure A25, Figure A26, Figure A27, Figure A28, Figure A29, Figure A30, Figure A31, Figure A32, Figure A33, Figure A34, Figure A35, Figure A36, Figure A37, Figure A38, Figure A39, Figure A40, Figure A41, Figure A42, Figure A43, Figure A44 and Figure A45 in Appendix A). All sentences referred to actions performed by anthropomorphic animals with masculine grammatical gender in Spanish. Anthropomorphic animals were chosen, rather than human characters, in order to control for grammatical gender and avoid potential differences in the perceived plausibility of each character performing each action (e.g., men vs. women, adults vs. children, etc.). Examples of the SRC and ORC sentences are presented in Table 3. While other syntactic formulations of relative clauses are possible in Spanish, the forms in (3) and (4) are acceptable and attested in natural speech.

Linguistic materials were recorded by a female native speaker of Spanish and presented auditorily to participants as they inspected a visual context containing three images (see Figure 2), each depicting the same three characters (e.g., a cat, a rabbit, and a monkey) interacting. We termed the temporal region in the spoken sentence preceding the relative clause the “anticipation region”. During the anticipation region, both types of sentences presented only a single NP that corresponded to one of the three displayed characters, and so in principle was consistent with all three images. However, as mentioned above, the three images differed with respect to the depicted agency of the initially mentioned NP. In the SRC condition (3), the referent of the mentioned NP (*el mono* “the monkey”) was performing two actions in the rightmost image, one action in the middle image, and no actions in the leftmost image. On the other hand, in the ORC condition (4), the referent of the mentioned NP (*el conejo* “the rabbit”) was performing two actions in the leftmost image, one action in the middle image, and no actions in the rightmost image.

The “relative clause region” extended from the onset of the relativizer *que* to the onset of the matrix verb. Comprehension of *que* as a relativizer signaled the listener that there would be an upcoming “gap” (marked with an underscore in Table 3) coindexed with the initial NP. In SRCs, this gap immediately followed *que*; in ORCs, this gap did not come until after the embedded verb. One competitor image (termed the “other RC” competitor) could be discarded after comprehension of the relative clause, but it was consistent with the *reverse* interpretation of the relative clause. That is, for an SRC stimulus, the “other RC” competitor was consistent with an interpretation of the matrix subject as the object of the relative clause, while for an ORC stimulus, this image was consistent with an interpretation of the matrix subject as the subject of the relative clause. The other competitor image (termed the “consistent” competitor) remained consistent with the linguistic input throughout the relative clause region. However, comprehension of the syntactic structure of the sentence up to this point would be expected to cause participants to disprefer the “consistent” competitor. The structure of the sentence until the end of the relative clause entails that the initial NP will be the subject of an upcoming matrix verb, which is inconsistent with the “consistent” competitor. In this way, basic syntactic comprehension of the relative clause would be indicated by decreased fixations to the “other RC” competitor, while more detailed syntactic structure building would be indicated by decreased fixations to the “consistent” competitor. After comprehension of the matrix clause, only the target image was consistent with the linguistic input. Examples of each image type, in correspondence with the linguistic stimuli in Table 3, are displayed in Figure 2.

In addition to the experimental manipulations of relative clause type and image type, our design integrated the composite measure of language dominance generated by the BLP as a continuous variable. A single repeated-measures experiment was presented to every participant containing both the SRC and ORC versions of each item, fully randomized by participant.

### 2.3. Procedure

Participants read and signed an informed consent form before the experiment began. Their eye movements were recorded at a sampling rate of 60 Hz using a Tobii TX300 eye-tracker as they inspected a visual display and listened to linguistic materials. Stimuli were presented with E-Prime 2.0 [53]. During each trial, participants first saw a cross on the computer screen. After they clicked on it, the cross disappeared, and a set of three images appeared (as in Figure 2). The relative positions of the images were randomized by trial. After two seconds, another cross appeared centered above the images. Participants were instructed to click this cross once they were familiar with the images in order to listen to the spoken sentence. When participants clicked on the second cross, it disappeared, and the auditory stimulus began playing while the three images remained on the screen. At the beginning of the experiment, participants were instructed to click the image that best represented what each spoken sentence conveyed. Then they completed five practice trials with stimuli unrelated to the experimental stimuli (see Figure A1, Figure A2, Figure A3, Figure A4 and Figure A5 in Appendix A). After the first practice trial, participants were told the correct answer and given an explanation for why that image was correct. After the fifth practice trial, participants were instructed to ask the experimenter any questions they had. Accuracy data were not collected during the practice trials.

### 2.4. Analysis

We analyzed offline comprehension accuracy using a logistic mixed-effects model conducted in R [54] with fixed effects of sentence type (sum-coded), language dominance as a continuous variable, and their interaction, and maximal by-subject and by-item random effects. In order to analyze the primary dependent variable of gaze fixation, three areas of interest (AOI) corresponding to the three displayed images were first defined using E-Prime 2.0 [53]. We then used R to inspect every gaze sample for every participant and trial. When a participant fixated on a given AOI, it was assigned a value of 1; otherwise, it was assigned a value of 0. Subsequently, we aggregated the samples into short time bins by calculating the mean fixation proportion for every 50 milliseconds by participant, trial, sentence type, and AOI. Finally, we calculated the mean fixation proportion by participant and the corresponding within-subject 95% confidence intervals (95% CI, see [55]) for each AOI on each sentence type for each 50 ms time bin.

We divided the gaze data into two time windows of interest. The first (the anticipation region) was a 1400 ms time window ending at the onset of the relativizer *que*. A duration of 1400 ms was chosen in order to approximate the mean onset time of the sentence across stimuli so that this region stretches from the approximate beginning of the stimulus to the exact onset of the relative clause. The second time window (the relative clause region) extended for 1900 ms from the exact onset of the relativizer. Again, a duration of 1900 ms was chosen in order to approximate the mean distance between the onset of *que* and the onset of the matrix verb, i.e., the duration of the relative clause. Inferential analysis was carried out using a quasi-logistic multilevel growth curve analysis (GCA) approach [56,57] on an empirical logit transformation of the proportion of fixations [58]. This statistical approach explicitly integrates time as a continuous variable into a single analysis, preventing multiple comparisons and power loss. The GCA uses orthogonal higher-order polynomials as predictors for non-linear changes on the dependent variable over time that characterize the interaction of visual attention and language processing.

The GCA models included the main effects of sentence type, image type, and language dominance as a continuous variable, as well as the interactions between these three factors. In the GCA model of the anticipation region, the images were coded based on the depicted agency of the mentioned NP (two-action, one-action, no-action). In the GCA model of the relative clause region, the images were coded based on the syntactic interpretation they represented (target, “consistent” competitor, “other RC” competitor). In both models, the sentence type factor was sum-coded, and the image type factor was treatment-coded (with “two-action” or “target” as the intercept). In Section 3, significant effects of language dominance are presented in a subsection to ease interpretation, although they pertain to the same statistical model. Following [56], the number of polynomial predictors was determined via model comparison. All models included the polynomial term(s) as main effects, as well as the interactions between each polynomial term and the three experimental factors described above. The polynomial terms and their interactions were included to improve the accuracy of the models, but they will not be discussed in the main text of the paper since they do not directly bear on our research questions. Full model summaries can be found in Table A1 and Table A2 in Appendix A. The random structure of the models included crossed random intercepts for participants and items, as well as random slopes for each polynomial term. In order to ease convergence, no random correlations between random effects were specified [59]. Following [60], we considered all effects where |*t*|> 2 as significant.

## 3. Results

A summary of participants’ offline comprehension accuracy is presented in Table 4, and the results of the logistic mixed-effects model are presented in Table 5. While accuracy was generally very high, participants were significantly less accurate on ORCs than SRCs. Neither language dominance nor the interaction between sentence type and language dominance were significant predictors of accuracy.

Having seen that participants were generally very accurate in the task and that language dominance had no significant effect on offline behavioral performance, we now turn to the eye-tracking data, which constitutes the primary dependent measure. The following two subsections present the GCA analysis of the gaze data in the anticipation region, followed by the GCA analysis of the gaze data in the relative clause region. In the presentation of each analysis, language dominance effects are reported in a separate subsection. In Section 4, the results are summarized and interpreted with respect to our research questions.

### 3.1. GCA: Anticipation Region

The results of the GCA analysis of the anticipation region are presented in Figure 3 and in Table A1 in Appendix A. During SRCs, looks to the no-action image showed the most subdued increase throughout the region. Looks to the one-action image initially showed the most rapid increase, but this increase began to subside about 1000 ms before the end of the region, before leveling off about 750 ms before the end of the region. Looks to the two-action image, however, continued to increase steadily until about 500 ms before the end of the region, and this image continued to be preferred over the others through the end of the region. While the gaze patterns during ORCs ultimately reflected the same preferences observed in SRCs, i.e., the two-action image was preferred over the one-action image, which was preferred over the no-action image, participants generally took much longer to evidence this preference in ORCs compared to SRCs. That is, in ORCs, the no-action image was preferred over both other images until about 500 ms before the end of the region, and it was preferred over the one-action image until about 250 ms before the end of the region.

As seen in Table A1, significant main effects of image type (one-action and no-action) confirm the general tendency that the two-action image was preferred over the one-action image and the no-action image, suggesting that participants generally preferred images that depicted greater agency of the referent of the mentioned NP, consistent with existing literature on event cognition. Interestingly, the agency preference we observed was scalar, in that the two-action image was preferred over the one-action image. A significant main effect of sentence type, as well as significant interactions between sentence type and image type (one-action and no-action), reflect the influence of sentence type on this scalar agency preference: in SRCs compared to ORCs, the preference was stronger for the two-action image and the one-action image, but weaker for the no-action image. Given that, in both conditions, participants heard only a single NP in this region, why would the preference for the agency of this NP be stronger in the SRC condition than in the ORC condition? A subtle difference in the relationship of the mentioned NP to the one-action image may offer an explanation of this by-condition difference. As seen in Figure 2, in ORCs, the one-action image also depicted the mentioned character (*el conejo* “the rabbit”) as the *patient* of an action, while in SRCs, the mentioned character (*el mono* “the monkey”) was only an agent and not a patient in this image. Therefore, the clearer pattern of agency preference observed in SRCs compared to ORCs might be attributable to the fact that ORC gaze patterns were additionally influenced by a *dispreference* for patiency, distinct from the hypothesized preference for agency.

#### Language Dominance Effects in the Anticipation Region

In order to address our third research question (see Section 1), we present in Figure 4 a plot of fixation proportions divided into 50 ms time windows and aggregated by image, sentence type, and participant (ordered by language dominance score). This allows a visual representation of the continuous effects of language dominance on fixation proportions toward each image, as well as the way that those effects unfold in time.

As seen in Figure 4, in SRCs, the observed mid-region attenuation of the preference for the two-action image was magnified by greater Spanish dominance until about 200 ms before the end of the region; and during a similar time window, greater Spanish dominance increased the preference for the no-action image; while looks to the one-action image were largely unaffected by Spanish dominance until the end of the region. In ORCs, from about 800 to 100 ms before the end of the region, greater Spanish dominance corresponded with an increase in looks to the one-action image and a decrease in looks to the two-action image, consistent with the general trend that Spanish dominance decreased the preference to interpret the mentioned NP as an agent. Interestingly, greater Spanish dominance also corresponded with an early *decrease* in looks to the zero-action image, which is inconsistent with this general trend. Given that it began very early in the region (approximately 300 ms after the average onset of the NP, earlier than any other observed effect), it is difficult to interpret in light of linguistic processing preferences. As seen in Table A1, significant interactions between dominance and image type (one-action and no-action) confirm that, across conditions, greater Spanish dominance weakened the preference for the two-action image over the one-action and no-action images. Significant three-way interactions between image type (one-action and no-action), sentence type, and dominance reveal the influence of sentence type: while greater Spanish dominance corresponded with a decrease in looks to the two-action image during both SRCs and ORCs, the positive effect of dominance on looks to the no-action image was mainly observed during SRCs, and the positive effect of dominance on looks to the one-action image was mainly observed during ORCs.

### 3.2. GCA: Relative Clause Region

The results of inferential analysis for the relative clause region are presented in Figure 5 and Table A2. As seen in Figure 5, at the beginning of the relative clause region, gaze patterns were consistent with the trends from the anticipation region. In both SRCs and ORCs, the proportion of fixations to the two-action image began higher than that to the one-action image, which began higher than that to the no-action image. In SRCs, these relative proportions remained approximately steady until about 500 ms after the onset of the relativizer, suggesting that the agency preference observed during the anticipation region was no longer actively altering fixation trajectories during this initial period of the relative clause region. Approximately 500 ms following the onset of the relativizer, while looks to the target (two-action) image continued to remain approximately steady, looks to the competitors diverged: looks to the “consistent” competitor (one-action) image rose, while looks to the “other RC” competitor (no-action) image fell. As described in Section 2.2, this pattern likely reflects the beginning of syntactic comprehension of the relative clause, as the “other RC” competitor was the only image that was inconsistent with the information received during this region. Given that the duration of saccade planning is approximately 200 ms [61,62], the time course of this pattern suggests that participants began to prefer an SRC structure within approximately 300 ms of the onset of *que*. Recall, in addition, that while the perception of *que* signals the onset of a relative clause, it is the word following *que* that conveys the information necessary to disambiguate between an upcoming SRC and ORC. Since the average onset of this word was only approximately 160 ms after the onset of *que*, i.e., before the 300 ms mark at which saccade planning is hypothesized to have begun, this pattern does not constitute unambiguous evidence for prediction. However, the very rapid onset of this pattern is compatible with at least an early syntactic *preference* for SRCs, such that SRC structures were preferred upon recognition of an upcoming relative clause. This pattern persisted until about 1000 ms after relativizer onset, when looks to the target and “consistent” competitor began to diverge, suggesting that participants were beginning to anticipate the syntactic structure of the linguistic information that would follow the offset of the relative clause (as described in Section 2.2).

In ORCs, after the onset of *que*, looks to the “other RC” competitor (two-action) image continued to rise rapidly, while looks to the “consistent” competitor (no-action) image continued to fall rapidly. This pattern contrasts with that observed in SRCs, where fixation trajectories to the three images were mostly steady for the first 500 ms of this region, without any rising or falling. There are two possible explanations of this by-condition difference, which are not mutually exclusive. First, it is possible that the agency preference observed in the anticipation region persisted longer in ORCs than SRCs, given that it began later (see Section 3.1). However, this pattern is also consistent with a syntactic preference for SRCs, as the “other RC” competitor was the only image that was consistent with an interpretation in which the initially mentioned NP was subject-extracted. The present results are not able to empirically distinguish between these two explanations; however, given that an SRC preference, beyond the agency preference, was observed in the SRC condition, it is likely that a similar syntactically-conditioned preference played a role during ORC processing. A qualitative shift in fixation trajectories was observed at about 700 ms after relativizer onset, as looks to the “other RC” competitor began to fall while looks to the “consistent” competitor began to rise. This suggests a re-analysis, as participants comprehended the word following *que* (in this case, another NP), which was inconsistent with an SRC interpretation. Following this re-analysis period (at about 1250 ms following relativizer onset), looks to the “other RC” competitor continued to fall, consistent with basic comprehension, while looks to the target began to rise more rapidly than looks to the “consistent” competitor, indicating preemptive syntactic structure building.

As seen in Table A2, main effects of image type (both competitors) and sentence type show that fixation proportions to the target were overall larger than fixation proportions to either competitor, as well as larger for SRC sentences compared to ORC sentences. Significant two-way interactions between image type (both competitors) and sentence type indicate that the difference in fixation proportions between the target and each competitor was modulated by sentence type. For SRC sentences, the preference for the target over the “consistent” competitor was smaller than for ORC sentences, while the difference between the target and the “other RC” competitor was larger for SRC sentences compared to ORC sentences. Recall that the “consistent” competitor represented an SRC interpretation in the SRC condition but an ORC interpretation in the ORC condition, and vice versa for the “other RC” competitor. Therefore, these interactions are consistent with an SRC preference in both conditions. Overall, the results in the relative clause region are consistent with the well-attested SRC/ORC processing asymmetry and suggest an apparent initial preference for an SRC interpretation even while perceiving an ORC.

#### Language Dominance Effects in the Relative Clause Region

As seen in Figure 6, in SRCs, greater Spanish dominance corresponded with a decrease in the preference for the “consistent” competitor that extended through most of the region. Until about 500 ms after relativizer onset, while overall fixation trajectories were still approximately steady, this trend was likely a continuation of the effect observed in the anticipation region. However, after about 1000 ms following relativizer onset, as overall looks to the “consistent” competitor began to fall, this trend likely reflects a facilitatory effect of dominance on comprehension, consistent with earlier findings [6,7] that greater dominance facilitates SRC processing. This interpretation is supported by the positive effect of Spanish dominance on target fixations beginning about 700 ms following relativizer onset. Moreover, while there was an early positive effect of Spanish dominance on looks to the “other RC” competitor (likely a continuation of the effect observed in the anticipation region), a late negative effect of Spanish dominance on looks to the “other RC” competitor also support an account in which greater Spanish dominance facilitated SRC processing.

In ORCs, both the early rise in looks to the “other RC” competitor and the early fall in looks to the “consistent” competitor were attenuated by greater Spanish dominance. That is, greater Spanish dominance corresponded with a decrease in looks to the “other RC” competitor and an increase in looks to the “consistent” competitor. This might be interpreted as further evidence for a facilitatory effect of dominance on syntactic processing, as comprehension of the relative clause should lead to a rejection of the “other RC” competitor. On the other hand, this pattern is also consistent with a continuation of the trend from the anticipation region, where greater Spanish dominance generally decreased the agency preference (recall that in the ORC condition, “other RC” competitor = two-action, and “consistent” competitor = no-action). The latter explanation seems to be supported by the fact that Spanish dominance was found to attenuate the trends later in the region: while overall looks to the “other RC” competitor fell, Spanish dominance increased looks to the “other RC” competitor; while overall looks to the target rose, Spanish dominance decreased looks to the target; and while overall looks to the “consistent” competitor fell, Spanish dominance increased looks to the “consistent” competitor. This later pattern suggests that greater Spanish dominance generally *decreased* processing speed during ORCs, slowing down rejection of the two competitor images and, correspondingly, convergence on the target.

The GCA summarized in Table A2 revealed a significant two-way interaction between language dominance and sentence type, indicating that greater Spanish dominance amplified the difference in target fixation proportions between sentence types. Another two-way interaction between dominance and image type (“consistent” competitor) indicates that, across sentence types, greater Spanish dominance corresponded with a larger difference in fixation proportions between the target and the “consistent” competitor. However, significant three-way interactions between dominance, image type (both competitors), and sentence type indicate that this effect was modulated by sentence type: greater Spanish dominance corresponded with a larger difference between target fixations and competitor fixations for SRC sentences, but a smaller difference between target and competitor fixations for ORC sentences. Overall, increased Spanish dominance corresponded with an increase in the SRC/ORC processing asymmetry. As mentioned above, both the beneficial effect on SRC processing and the negative effect on ORC processing can be explained by an increased expectation for an SRC at the onset of the relative clause [6,7].

A summary of the main findings is presented in Table 6. Regarding research questions 1 and 2 (see Section 1), participants demonstrated both a syntactic preference for SRCs over ORCs in the relative clause region (SRC target > ORC target), as well as a semantic preference for agency in the anticipation region (two-action image > one-action image > no-action image), consistent with our hypotheses. Regarding research question 3, we observed that greater Spanish dominance increased looks to the target and decreased looks to the competitors during SRC stimuli and vice versa during ORC stimuli. We interpret this as evidence that greater Spanish dominance magnified the SRC preference, consistent with our hypothesis. As stated in Section 1, we had no specific prediction regarding the effect of language dominance on the agency preference in the anticipation region. We observed that greater Spanish dominance decreased looks to the two-action image and increased looks to the one-action and no-action images, consistent with an attenuative effect of dominance on the agency preference.

## 4. Discussion

The eye-tracking results indicated two distinct influences on participants’ processing of the linguistic stimuli: a semantically driven preference to assign agency to perceived NPs, and a syntactically driven preference to interpret relative clauses as subject-extracted. Moreover, greater language dominance was found to correspond with a decrease in the semantic preference for agency and an increase in the syntactic preference for SRCs.

In the anticipation region, upon the sole mention of an NP, images depicting the referent of this NP as a thematic agent were preferred over images depicting this referent as a patient. Moreover, this agency preference manifested in a scalar way, such that the two-action image (e.g., the mentioned NP biting and grabbing) attracted more looks than the one-action image. Lastly, the by-condition difference observed in this processing pattern points to a possible distinction between the agency preference and a patiency dispreference in processing.

Since, in the anticipation region, participants received no syntactic information (beyond a single NP), the observed pattern was likely generated by semantic proclivities to assign or expect agency in event construction (and to *not* expect patiency). That is, upon hearing an (animate) NP, the expectation is for the referent of that NP to act rather than to be acted upon. This result is consistent with findings on the cognition of event structure where psycholinguistic experiments, visual recognition studies, infant perception studies, and home sign language studies have shown the primacy of assigning agent roles over patient roles [28,29].

In the relative clause region, beginning with the relative pronoun *que*, images depicting an SRC interpretation were preferred over those depicting an ORC interpretation. It is possible that this apparently syntactic preference was, in fact, driven by the semantic preference observed in the anticipation region. A long-standing position in linguistics holds that thematic roles project to functional positions in the sentence, such that agents instantiate as subjects while patients instantiate as objects [63,64]. While this has mainly been postulated for main clauses, it is possible, indeed likely, that this projection would extend to subordinate clauses as well. In our case, upon mention of the relative pronoun *que*, syntactic structure building can begin for a relative clause. However, this process is likely to be influenced by the thematic role expectations originating in the anticipation region, thus maintaining agency for the NP and assigning a subject position to that NP in the subordinate clause. If so, semantic event structure expectations can be said to drive syntactic structure building, favoring SRCs and disfavoring ORCs. The convergence of semantic and syntactic forces might then explain the subject/object asymmetry seen in the relative clause region.

However, an explanation in which the semantic agency preference and the syntactic SRC preference are entirely interdependent is complicated by the fact that language dominance exerted *opposite* effects on each of these preferences. That is, greater dominance decreased the agency preference but increased the SRC preference. If these two preferences were, in fact, reducible to a single source, it would be puzzling (if not contradictory) that an individual-level factor such as language dominance could decrease one while increasing the other. Therefore, although it is clear that these two preferences often cause converging processing patterns (as described above), it is likely that they emerge from (at least somewhat) distinct sources.

Although this characterization is admittedly vague, a more detailed proposal regarding the mechanisms underlying the observed relationships between dominance and processing would, at this point, be speculative. Nonetheless, some reasoned speculation might be useful to the extent that it can drive future research. On the one hand, it is possible that the observed negative relationship between language dominance and the agency preference is a consequence of the particular languages spoken by the participants in this study: Spanish, participants’ L1 and the language of the experiment, and English, participants’ L2. It has been argued, for example, that Spanish has seen an increase in certain constructions, such as the *se*-construction, left-dislocation, and plural impersonal, whose function is to highlight an affected entity (i.e., a prototypical patient, as in the middle construction “the vase broke”) while strongly de-emphasizing the agent, often to the point of omission [65]. It is possible, then, that the negative effect of Spanish dominance on the agency preference could be particular to Spanish, perhaps from the frequency of such constructions in this language. Of course, this is merely a conjecture, but our point is that there is likely to be variability across languages in the degree to which they instantiate the agency preference.

On the other hand, the negative effect of dominance on the agency preference also suggests a more general interpretation: perhaps there is a trading relation between semantic and syntactic processing strategies during sentence processing. In the case of relative clause processing, perhaps greater reliance on the syntactic preference to interpret the structure of the clause as subject-extracted corresponds with a reduced reliance on the semantic preference to assign agency to NPs. According to this view, to the extent that increased dominance tends to increase the syntactic SRC preference [6,7], it would be expected for dominance to correspondingly decrease the semantic agency preference. If we further assume that syntactic structure building strategies are more language-specific than event cognition strategies, then this explanation is intuitive in the sense that decreased language dominance would be expected to decrease syntactic structure building strategies while having little to no attenuating effect on general event cognition strategies.

As stated above, this account is mostly speculative at this point. However, it makes testable predictions. For instance, this account predicts that a similar dominance-mediated trading relation between semantic and syntactic processing strategies should be observed in later-learned languages in addition to the L1. In fact, some evidence in support of this prediction has been reported in previous studies. Although the studies of which we are aware have relied on group comparisons, rather than individual-level measures, a body of work has demonstrated that bilinguals and L2 learners use morphosyntactic cues during processing to a reduced degree compared to monolingual counterparts [66,67,68,69,70], while at least one previous study has demonstrated evidence that bilinguals exhibit *increased* semantic prediction relative to monolingual counterparts [71]. Of course, more research is needed to understand the relationship between semantic and syntactic processing in later-learned languages, particularly regarding how this relationship is modulated by continuous factors at the level of the individual. Moreover, even in monolinguals, there might be individual variation in the extent of reliance on syntactic and semantic processing strategies. If our speculation is on the right track, then this variation should be structured such that there should be a negative correlation between indices of semantic and syntactic processing strategies.

Of course, other dimensions of individual variation are also likely to play a role in shaping the relationship between semantic and syntactic processing strategies. Individual differences in cognitive control, for instance, have been found to modulate the resolution of competing semantic and syntactic cues during thematic role assignment, such that comprehenders with greater cognitive control converge on the correct interpretation more quickly [70]. Regarding relative clause processing, it is thus possible that individuals with greater cognitive control would be able to more quickly resolve competition between the (semantic) agency preference and the (syntactic) SRC preference via more rapid selective inhibition of misleading cues. Investigating this possibility would be a fruitful area for future research. Similarly, individual differences in working memory have been found to modulate parsing preferences during the comprehension of syntactically ambiguous relative clauses [32,72,73]. It would thus be useful to investigate the possible influence of working memory on the relationship between the agency preference and the SRC preference during relative clause processing (see Section 1). Finally, it is important to point out that the conceptualization of language dominance used in the present study is essentially *symmetrical*, in that a score of, say, 50 (relatively Spanish-dominant) has the same meaning regardless of whether Spanish is the L1 or the L2 of the bilingual participant. However, it is likely that order of acquisition plays a role independently of relative dominance at the time of testing. While some work has compared the role of multidimensional language dominance in L1 and L2 processing [74], more work is needed to understand the potential asymmetricality of language dominance effects.

The results of the present study demonstrate the utility of gradient, multidimensional, individual-level measures in understanding bilingual language processing. As evidenced by self-ratings of proficiency (Table 1) and offline accuracy scores on the experimental task (Table 4), all participants in the present study were highly proficient in their L1 (Spanish), the language of the experimental task. However, through the more granular measure of language dominance, in combination with the moment-by-moment tracking of processing via visual world eye-tracking, we were able to detect subtle patterns in the relationship between bilingual experience and L1 processing. Crucially, some of the patterns we observed, particularly among the more English-dominant participants, differed from the patterns observed in monolingual speakers of both Spanish *and English*, so they cannot be explained solely via normative comparison to monolinguals. Moreover, it is not clear that any of the processing patterns we observed (e.g., greater reliance on syntactic versus semantic preferences) was “better” than any other, since all patterns entailed both benefits and detriments with regard to processing time, depending on the type of linguistic stimulus being comprehended. Broadly, the present study highlights the complexity of the relationship between the representation of multiple languages in the mind and the processing of those languages in real time, and the importance of avoiding reductionist explanations that rely solely on normative comparison to monolinguals.

## 5. Conclusions

In this study, we investigated the effects of language dominance on the subject/object relative clause processing asymmetry in the first-learned language (Spanish) of highly proficient Spanish-English bilingual adults. We treated both language dominance and time as continuous variables in order to pursue a fine-grained understanding of the relationship between individual-level language dominance and the time course of relative clause processing. In particular, we examined gaze fixation patterns before the onset of disambiguating linguistic information in order to probe the effects of language dominance on predictive processing. We found that, upon hearing an initial NP, participants exhibited a predictive processing strategy based on thematic role assignment, such that images depicting greater agency of the mentioned NP were preferred. After the onset of the relative pronoun *que,* which signaled the onset of a relative clause, participants exhibited a preference for images depicting an SRC interpretation rather than an ORC interpretation, suggesting an online syntactic preference to interpret relative clauses as subject-extracted. Although it is possible that the apparently syntactic preference for SRCs was, in fact, entirely driven by the semantic preference for agency, we argued that these two preferences are likely distinct since language dominance was found to exert opposite influences on each of them. That is, dominance decreased the semantic preference for agency, but increased the syntactic preference for SRCs. We speculated that these dual influences might be attributable to a trading relation between semantic and syntactic processing strategies, such that as dominance increases reliance on syntactic processing strategies, there is a corresponding decrease in reliance on semantic processing strategies. We suggested paths for future research to test the predictions of this speculation.

## Figures and Tables

**Figure 1 brainsci-11-00989-f001:**
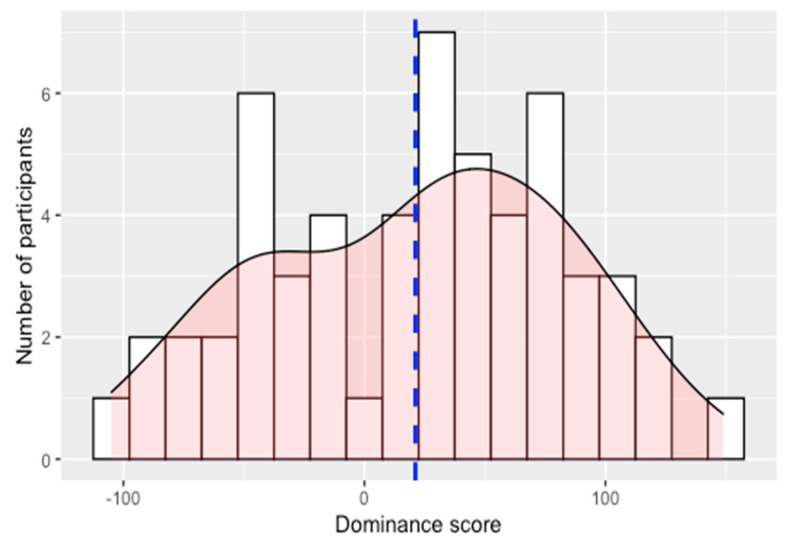
Histogram (bin width = 15) and density plot of participants’ dominance scores. Positive scores indicate Spanish dominance, and negative scores indicate English dominance. The vertical dotted line indicates the mean.

**Figure 2 brainsci-11-00989-f002:**
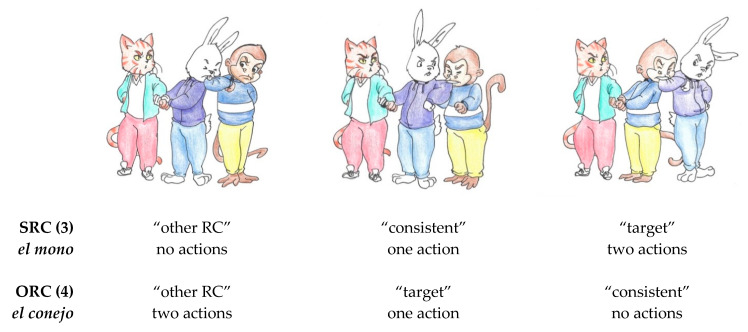
Example materials for an experimental visual context corresponding to (3) and (4).

**Figure 3 brainsci-11-00989-f003:**
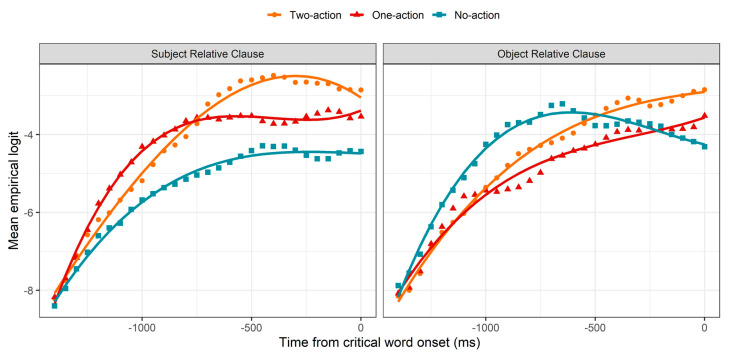
GCA model fit (lines) of empirical logit (points) as a function of image in the visual context and sentence type for the anticipation region.

**Figure 4 brainsci-11-00989-f004:**
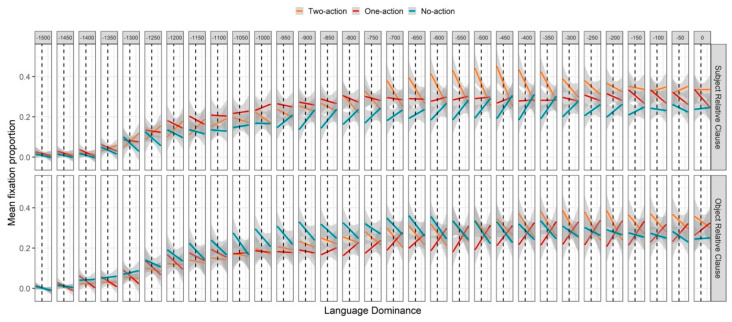
Mean fixation proportions in each 50 ms time window across the anticipation region, as a function of image type, sentence type, and language dominance, where positive values represent Spanish dominance, and negative values represent English dominance. The vertical dashed lines index a dominance score of zero (by hypothesis, balanced bilingualism). Orange lines represent fixation proportions toward the two-action image, red lines represent fixation proportions toward the one-action image, and turquoise lines represent fixation proportions toward the no-action image, while line slopes depict the effects of language dominance. Gray shading around the lines represents the standard error of the mean for the linear effect calculated for each individual 50 ms time window.

**Figure 5 brainsci-11-00989-f005:**
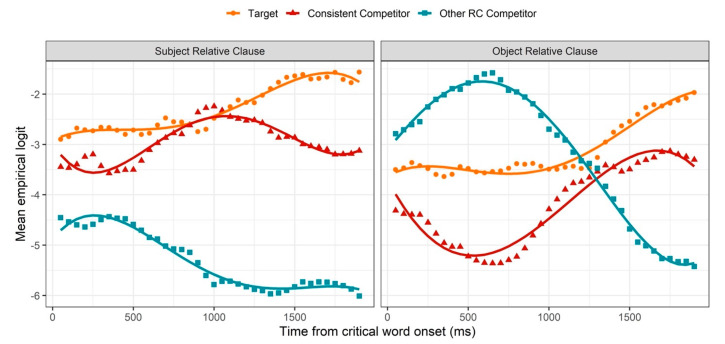
GCA model fit (lines) of empirical logit (points) as a function of image in the visual context and sentence type for the relative clause region.

**Figure 6 brainsci-11-00989-f006:**
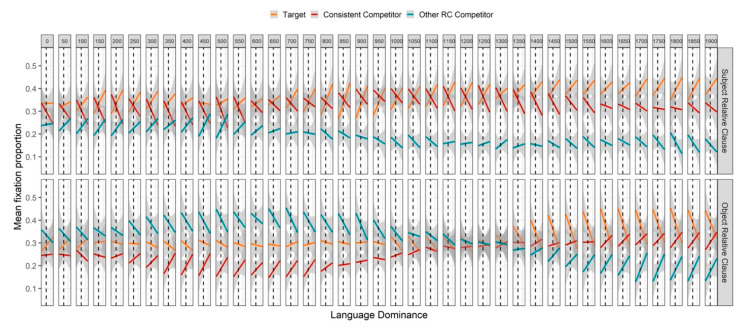
Same as Figure 4, but for the relative clause region.

**Table 1 brainsci-11-00989-t001:** Summary of participant characteristics. For the sake of illustration, participants are divided into the groups “Spanish-dominant” and “English-dominant” based on a zero split; however, there was no group variable in any of our inferential analyses.

Variable	Group (Zero-Split)	Mean	SD	95% CI (Lower)	95% CI (Upper)	Min	Max
Composite dominance score	Spanish-dominant	62.004	35.208	49.909	74.098	8.716	148.838
	English-dominant	−47.016	29.111	−60.267	−33.765	−104.980	−1.820
History module score	Spanish-dominant	26.929	11.091	23.119	30.739	−1.816	45.400
	English-dominant	−5.253	10.717	−10.132	−0.375	−29.510	9.080
Use module score	Spanish-dominant	7.381	16.518	1.707	13.055	−19.630	45.780
	English-dominant	−31.818	10.717	−36.696	−26.940	−47.960	−10.900
Proficiency module score	Spanish-dominant	7.134	7.886	4.426	9.843	−6.810	22.700
	English-dominant	−10.810	8.730	−14.783	−6.836	−27.240	0.000
Attitudes module score	Spanish-dominant	20.560	14.122	15.709	25.411	−13.620	45.400
	English-dominant	0.865	8.598	−3.049	4.779	−13.620	22.700
Age	Spanish-dominant	26.400	9.050	23.291	29.509	19	55
	English-dominant	30.095	8.927	26.032	34.159	20	48
Age of arrival to anglophone U.S.	Spanish-dominant	19.114	9.055	16.004	22.225	3	45
	English-dominant	5.190	6.853	2.071	8.310	0	25
Years lived in anglophone U.S.	Spanish-dominant	7.286	6.662	4.997	9.574	0	31
	English-dominant	24.905	8.479	21.045	28.764	6	42
Self-rated Spanish proficiency	Spanish-dominant	4.929	0.247	4.844	5.013	4	5
	English-dominant	4.500	0.671	4.195	4.805	3	5
Self-rated English proficiency	Spanish-dominant	4.200	0.740	3.946	4.454	3	5
	English-dominant	4.905	0.301	4.768	5.042	4	5

**Table 2 brainsci-11-00989-t002:** Summary of participants’ countries of birth.

	Number of Participants	
Country of Birth	Spanish-Dominant	English-Dominant
Argentina	1	0
Colombia	3	4
Cuba	0	1
Dominican Republic	11	3
Ecuador	6	0
Guatemala	0	1
Honduras	3	0
Mexico	4	0
Peru	2	0
Puerto Rico	1	1
Uruguay	1	0
USA (mainland)	1	10
Venezuela	2	1

**Table 3 brainsci-11-00989-t003:** Example stimuli from the SRC (3) and ORC (4) conditions.

	Anticipation Region	Relative Clause Region	Matrix Clause Region
(3)	El mono,	que __ muerde al conejo,	agarra al gato.
	the.M monkey	that bite.3SG to-the.M rabbit,	grab.3SG to-the.M cat.
	“The monkey,	that __ bites the rabbit,	grabs the cat”.
(4)	El conejo,	que el mono muerde __,	agarra al gato.
	the.M rabbit	that the.M monkey bite.3SG,	grab.3SG to-the.M cat.
	“The rabbit,	that the monkey bites __,	grabs the cat”.

**Table 4 brainsci-11-00989-t004:** Offline comprehension accuracy rate by sentence type.

Sentence Type	Mean	SD	95% CI (Lower)	95% CI (Upper)	Min (by Participant)	Max (by Participant)
SRC	0.936	0.325	0.909	0.963	0.4	1.0
ORC	0.850	0.466	0.811	0.889	0.4	1.0

**Table 5 brainsci-11-00989-t005:** Results of the logistic mixed-effects model of offline comprehension accuracy.

	Estimate	se	*z*-Value	*p*-Value
(Intercept)	2.967	0.262	11.321	<0.001
Dominance	0.001	0.004	0.230	0.818
Sentence Type	0.542	0.204	2.660	<0.01
Dominance: Sentence Type	0.004	0.003	1.322	0.186

**Table 6 brainsci-11-00989-t006:** Summary of main findings from each region, including main effects of predictors (Sentence Type, Image Type, Language Dominance) and their interactions. In the relative clause region, effects that are apparent continuations of effects from the anticipation region are not reported. The greater-than (>) and less-than (<) signs indicate inequalities between levels of a factor, and the minus sign (–) represents the difference between levels of a factor. Upward (**↑**) and downward (↓) arrows indicate increases and decreases, respectively, of the variable to their right. In the anticipation region, the timestamps between parentheses reflect milliseconds before the onset of the relativizer *que*. In the relative clause region, they reflect milliseconds after the onset of the relativizer.

Region	Predictor	Effects on the Proportion of Fixations	
Anticipation region	Sentence Type (SRC vs. ORC) and Image Type (two-action, one-action, no-action)	Overall, two-action image > one-action > no-action In SRCs compared to ORCs, ↑ two-action, ↑ one-action, ↓ no-action	
	Language Dominance	In SRCs, ↑ Spanish Dominance =	In ORCs, ↑ Spanish Dominance =
		↓ two-action (−800 to −200 ms)	↓ two-action (−800 to −100 ms)
		↑ no-action (−900 to −200 ms)	↑ one-action (−800 to −100 ms)
Relative clause region	Sentence Type (SRC vs. ORC) and Image Type (target, “consistent” competitor, “other RC” competitor)	SRC target > ORC target Target−“consistent” competitor: SRC < ORC Target−“other RC” competitor: SRC > ORC	
	Language Dominance	In SRCs, ↑ Spanish Dominance =	In ORCs, ↑ Spanish Dominance =
		↑ target (from 700 ms)	↓ target (from 1000 ms)
		↓ “consistent” competitor (1000 to 1600 ms)	↑ “consistent” competitor (from 1600 ms)
		↓ “other RC” competitor (from 1500 ms)	↑ “other RC” competitor (from 1300 ms)

## Data Availability

The data presented in this study are openly available on OSF at https://osf.io/prj8a/ (accessed on 27 July 2021).

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
