# Peer review of "Syntactic and Semantic Influences on the Time Course of Relative Clause Processing: The Role of Language Dominance"

_brainsci, 2021, doi:10.3390/brainsci11080989_

Round 1

Reviewer 1 Report

In this study, the authors studied a group of Spanish-English bilinguals using eye-tracking method and examined the effect of language dominance on the time course of relative clause processing (Subject Relative Clause and Object Relative Clause) in their first language (i.e., Spanish). The results showed two patterns: 1) semantically-driven preference to assign agency to referents of noun phrases and syntactically-driven preference to interpret the relative clause as subject-extracted; 2) greater dominance in Spanish led to greater reliance on syntactically-driven preference while lesser reliance on semantically-driven preference. Overall, the results point towards a positive relationship between language dominance and language-specific syntactic processing strategies and a negative relationship between language dominance and domain-general semantic processing strategies.

This manuscript can make a significant contribution to the relatively scant literature on sentence processing in the bilingual population. The results are interesting. However, the manuscript is too big and it was very difficult to follow the results. Therefore, I think the manuscript requires a major revision (especially the structuring of the manuscript) . I think restructuring the introduction and succinctly presenting the results would improve the readability of the manuscript immensely. I will highlight my concerns below: 

  1. The authors start the introduction with a framework of bilingual versus monolingual. However, this is not the aim of the study, and it gives a false sense to the readers of what they are about to expect. I think the introduction may start with explaining the relative clause processing in general (SRC versus ORC) and how language dominance modulates such processing. There is too much focus on language dominance and how it can be measured, etc. However, the goal of the study is not to compare different language dominance measurement methods.
  2. The authors have listed two primary research questions but did not provide any specific hypothesis in section 1.3! I think that can cause difficulty in understanding and interpreting the results. All the literature that the authors discuss in the discussion should be in the intro and then it would make sense to interpret the results. In this draft, the introduction and discussion sections are not in sync.
  3. I think a summary table of all the results (especially the result section that’s related to language dominance) would be beneficial to the reader in understanding the complex nature of all different kinds of analysis. I don’t think Figure 5, 7, 10 improve the readability of the results.
  4. I understand the authors did not measure the cognitive control abilities of the participants. However, it would be interesting to know whether differences in cognitive control abilities may influence the processing of relative clauses sentences (syntax specific versus domain-general). The authors may write few sentences to comment on that in the discussion section!

Reviewer 2 Report

Review Report Form

brainsci-1213649

Gradient effects of language dominance on the time course of relative clause processing in bilinguals’ first-learned language

In this paper, the authors aim to examine the effects of language dominance on L1 relative clause processing. It is an interesting study on the role of bilingualism in language processing. It provides a multidimensional way of measuring bilingualism and incorporates the treatment of the time course of L1 processing.

First of all, I will make some comments suggested to me by the approach of the study. Subsequently, I make suggestions that I think the authors should take into account when describing and discussing the results obtained.

Suggestions based on the approach of the study. To be taken into account in the discussion - future studies.

In the introduction, the authors rely on studies that have not yet been published. It is difficult to follow the approach taken since the study in question is not accessible.

An aspect that could also be considered concerning language dominance is that of language components (lexis, phonetics, morphosyntax, pragmatics). Bilinguals may differ in their proficiency in these aspects, due to aptitude, language use, etc. 

Studies on bilingualism must also consider the possibility of simultaneous bilingualism in which language dominance is also manifested. Thus, although the study focuses on L1a-L1b bilingualism, a comparison group could be that of simultaneous English-Spanish bilinguals (future studies).

Another possible comparison group would be L1-L2 bilinguals where language is English.

To be considered for the presentation of the results and especially for discussion.

No new data or results should be provided in the discussion section. The presentation of results and discussion should be restructured and organised differently. One possibility would be to have a partial discussion, new results (or another results section, study 2, subsequent study), and general discussion.

As the authors acknowledge, the processing of relative clauses has been extensively studied in several languages, in particular the differences in processing between subject and object relative clauses. Similarly, the role of animacy in preverbal sentences has also been discussed. However, in the case of bilingual subjects, the processing of different types of relative clauses in different languages may not be easily comparable.

In particular, in the case of relative clauses in English and Spanish, factors related to the frequency of use in one of these languages may be involved.

I have no information about the frequency of use of relative clauses in spoken or written Spanish, but it should be borne in mind that a sentence like "El conejo, que el mono muerde, agarra al gato" is not wrong, but neither is it natural in Spanish speech.

If it is used, it is more usual to introduce the clause with "al", as follows: "El conejo, al que el mono muerde, agarra al gato".

In short, it has long been noted that such a sentence, typical of processing experiments, will be more difficult to process than a subject relative sentence in both English and Spanish. However, there may also be differences in frequency of use between the languages under study.

Based on this observation, we must consider the possibility that the frequency and type of use of these sentences in Spanish compared to English is a determining factor in this study.
